# Exploring the One Health Perspective in Sweden’s Policies for Containing Antibiotic Resistance

**DOI:** 10.3390/antibiotics10050526

**Published:** 2021-05-03

**Authors:** Jaran Eriksen, Ingeborg Björkman, Marta Röing, Sabiha Y. Essack, Cecilia Stålsby Lundborg

**Affiliations:** 1Department of Global Public Health—Health Systems and Policy (HSP): Improving the Use of Medicines, Karolinska Institutet, Tomtebodavägen 18A, 171 77 Stockholm, Sweden; cecilia.stalsby.lundborg@ki.se; 2Unit of Infectious Diseases, Venhälsan, Södersjukhuset, 118 83 Stockholm, Sweden; 3Department of Public Health and Caring Sciences, Health Services Research, Uppsala University, Husargatan 3, 751 22 Uppsala, Sweden; ingeborg.bjorkman@nestorfou.se (I.B.); marta.roing@pubcare.uu.se (M.R.); 4Antimicrobial Research Unit, College of Health Sciences, University of KwaZulu-Natal, Durban 4001, South Africa; essacks@ukzn.ac.za

**Keywords:** One Health, Sweden, policy, policy analysis, antibiotic resistance

## Abstract

Antibiotic resistance is considered to be a major threat to global health. The main driver of antibiotic resistance is antibiotic use. Antibiotics are used in humans, animals, and food production and are released into the environment. Therefore, it is imperative to include all relevant sectors in the work to contain antibiotic resistance, i.e., a One Health approach. In this study, we aimed to describe and analyse Sweden’s policies related to containing antibiotic resistance, from a One Health perspective. Twenty-three key policy documents related to containment of antibiotic resistance in Sweden were selected and analysed according to the policy triangle framework. Sweden started early to introduce policies for containing antibiotic resistance from an international perspective. Systematic measures against antibiotic resistance were implemented in the 1980s, strengthened by the creation of Strama in 1995. The policies involve agencies and organisations from human and veterinary medicine, the environment, and food production. All actors have clear responsibilities in the work to contain antibiotic resistance with a focus on international collaboration, research, and innovation. Sweden aims to be a model country in the work to contain antibiotic resistance and has a strategy for achieving this through international cooperation through various fora, such as the EU, the UN system, and OECD.

## 1. Background

Antimicrobial resistance is a rapidly growing global challenge rendering previously easily treatable infections potentially fatal [1]. An estimated 10 million people are expected to die annually, by 2050, because of antimicrobial resistance without intervention [2]. This has led the 2016 UN General Assembly to conclude that antimicrobial resistance is one of the greatest threats to human health globally, and in need of urgent attention [3]. In this paper, we focus on antibiotics, the antimicrobials used against bacteria.

The main driver of antibiotic resistance is the use of antibiotics, and the rate of resistance development is related to the amount of antibiotics consumed [4,5,6]. In the World Health Organisation (WHO) Strategy for Containment of Antibiotic Resistance from 2001, there is a focus on a reduction in unnecessary and incorrect use of antibiotics as one of the key factors to slow the speed of resistance spread [7].

Antibiotics are not only used for human health. In fact, in many countries the majority of antibiotic consumption is by animals [8]. Furthermore, antibiotics are often given to healthy animals as growth promotors or for prophylactic and metaphylactic herd treatment, increasing the unnecessary use of antibiotics [6]. In addition, antibiotics have an impact on the environment. Due to incomplete metabolism of ingested antibiotics or disposal of unused antibiotics, antibiotics enter the environment [9,10]. In the environment, antibiotic residues can induce the development of antibiotic resistance genes in bacteria that can potentially be transferred back to animals and humans [9]. The realisation of this multifactorial cause of antibiotic resistance has led the global community to adopt the so-called “One Health” approach for containing antibiotic resistance. This approach is defined as a “collaborative, multisectoral, and transdisciplinary approach—working at local, regional, national, and global levels—to achieve optimal health and well-being of people, animals, plants and their shared environment, recognising the interconnections between them” [11].

This study is part of an international project, namely A One Health Systems and Policy Approach to Antibiotic Resistance Containment: Coordination, Accountability, Resourcing, Regulation and Ownership (ABR CARRO) which aims to explore and describe how national action plans on antibiotic resistance were developed, implemented, monitored, and evaluated in Sweden, South Africa, and Swaziland. The project includes interviews with different categories of stakeholders, at the government level, for example, policymakers and professionals in human, animal, and environment/agriculture sectors, as well as policy document analyses.

In this study, our focus is on Sweden, where work to contain antibiotic resistance in both the human and animal sectors was started relatively early and was intensified through the creation of the Swedish strategic programme against antibiotic resistance in the mid-1990s [12]. In this paper, we describe and analyse Sweden’s policies for containing antibiotic resistance, to contribute knowledge about the extent to which a One Health perspective is being incorporated into the Swedish policies, as well as to provide knowledge on strategies for containment of antibiotic resistance.

## 2. Methods

### 2.1. Study Setting

Sweden is a northern European high-income country with a population of about 10.3 million people [13] and low levels of antibiotic resistance [14]. The life expectancy is high, i.e., 81 years for men and 84 for women [13] and maternal mortality (4 per 100,000 births [15]) and child mortality (2.6 per 1000 live births [16]) are low. Healthcare in Sweden is mainly funded by taxes with little out-of-pocket costs for the patients. There are 21 Swedish Regions that are responsible for healthcare provision but overall healthcare policies are decided at the national level [17]. Antibiotic consumption in human medicine is decreasing in Sweden and is among the lowest in Europe [18,19]. Even sales of antibiotics per population corrected unit for veterinary use is low as compared with other European countries, and has decreased markedly since 1980 [19,20].

### 2.2. Framework

Policy analysis, and specifically health policy analysis, is a multidisciplinary approach to public policy that aims to explain the interactions among institutions, interests, and ideas in the policy process [21]. Policy analysis has been said to be crucial for evaluating the effectiveness of policies, and thereby aiding health sector reforms [22]. A number of policy analysis frameworks have been discussed and published, but for health policy analysis, the policy triangle framework by Walt and Gilson [22] has become established as one of the most commonly used frameworks. It is considered to be a broader framework which encompasses most factors included in other frameworks [23]. The policy triangle is used as a way to present the context of a policy, its content, the policy process, and the actors involved (Figure 1).

Researchers can use this framework to describe what has happened regarding a policy, and also include an explanation as to why it happened [24].

In our policy analysis, we used a One Health framework for antibiotic resistance and considered four policy analysis factors, as presented in Figure 1, i.e., context, content, actors, and process.

### 2.3. Data Collection

The data collection was first conducted during the second half of 2018, with an updated search in September 2020. Key policy documents were found by searching the relevant government agencies’ websites for literature regarding antibiotic use and resistance. Once we had identified documents, a list was compiled and sent to key stakeholders (experts at the Swedish Board of Agriculture, the Public Health Agency of Sweden, and the National Veterinary Institute) to ask if they knew of more relevant documents that could be added to our list. When we felt that we had reached saturation in this step, i.e., all new documents suggested were already on our list, a final list of documents to be analysed was compiled (see Appendix A).

We defined a policy document as a non-academic paper that articulated recommendations for interventions to be implemented, and that was provided by government actors and professional organisations [25].

### 2.4. Data Analysis

We selected 23 main policy documents for our analysis (see Appendix A). We read through the documents highlighting information according to the four policy analysis factors (content, context, process, and actors). This information was compiled in an excel sheet, making note of specific mentions of One Health. After a first review, five documents were selected for in-depth analysis. These documents were selected either because they were updated versions of older documents or of a more overarching character covering several topics that were covered separately in some of the other policy documents. Thereafter, the other documents were reviewed again with a special focus on information that complemented the key documents. The key documents that were analysed in-depth were as follows:Swedish strategy to combat antibiotic resistance 2020–2023 (Government Offices of Sweden, 2020);Revised multisectoral action plan against antibiotic resistance 2018–2020 (“Reviderad tvärsektoriell handlingsplan mot antibiotikaresistens 2018–2020”) (Public Health Agency of Sweden & Swedish Board of Agriculture, 2017);Action plan against antibiotic resistance and healthcare associated infections—background for the agencies’ continued work (“Handlingsplan mot antibiotikaresistens och vårdrelaterade infektioner—underlag för myndigheternas fortsatta arbete”) (National Board of Health and Welfare & Swedish Board of Agriculture, 2015);Swedish work on containment of antibiotic resistance (Public Health Agency of Sweden, 2014) (this document describes Sweden’s work rather than a policy, as the other documents do);Surveillance of antibiotic resistance—national plan 2014 (“Övervakning av antibiotikaresistens—nationell plan 2014”) (Public Health Agency of Sweden, 2014).

## 3. Results

The policies are described below according to the four main policy analysis factors.

### 3.1. Content

#### 3.1.1. Swedish Strategy to Combat Antibiotic Resistance 2020–2023

Based on the “Swedish strategy to combat antibiotic resistance” adopted in April 2016, this updated strategy was published in April 2020. It states an overarching goal of preserving the possibility of effective treatment of bacterial infections in humans and animals, and lists the following seven main objectives:Increased knowledge through enhanced surveillance;Continued strong preventive measures;Responsible use of antibiotics;Increased knowledge for preventing and managing bacterial infections and antibiotic resistance with new methods;Improved awareness and understanding in society of antibiotic resistance and countermeasures;Supporting structures and systems;Leadership within the EU and in international cooperation.

The strategy outlines antibiotic resistance as a problem of global dimensions that affects all. It also highlights the complexity of antibiotic resistance and its transmission between humans, animals, food, and the environment, and advocates the One Health approach. Containing antibiotic resistance requires intense global efforts. The strategy has a focus on Sweden’s international leadership in action against antibiotic resistance, aiming for continued engagement in advocacy and setting a good example. The document contains a section about Sweden as a model for the containment of antibiotic resistance.

#### 3.1.2. Revised Multisectoral Action Plan against Antibiotic Resistance 2018–2020

This is an updated version of the multisectoral action plan from 2015. The plan includes healthcare, public health, the environment, animal husbandry, veterinary medicine, food production, and research, but is limited to national agencies and work that requires cooperation between agencies and sectors. The action plan is meant as support for the so called “coordination mechanism” that aims to coordinate the multisectoral work against antibiotic resistance. This coordinating function consists of 25 agencies/organisations, led by the Public Health Agency of Sweden and the Swedish Board of Agriculture. The action plan describes the roles and areas of responsibility of each of the actors involved in the work. It lists very specific action points, such as “arrange yearly intersectoral meeting” and “plan and ensure funding for regular surveys to follow-up the cross-sectoral study of spread of ESBL(extended-spectrum beta-lactamase) carrying bacteria between humans, animals, food and environment”, and the responsible actor for each action.

#### 3.1.3. Action Plan against Antibiotic Resistance and Healthcare Associated Infections: Background for the Agencies’ Continued Work

The action plan lists the following six target areas:International work;Knowledge and competence;Prevent, detect, rectify;Wise and rational use of antibiotics;Data collection and analysis;Diagnostics and effective treatment.

Within each target area there are lists of pressing measures to be taken and the responsible agency or organisation. Target areas No. 1 and No. 2 are overarching areas, whereas the remaining four all have specific measures for the sectors human health, environment, and animal health/foodstuff. The plan also contains prerequisites for successful work, within human health, the environment, animal health/foodstuff, as well as research and innovation. The latter section mentions the Joint Programming Initiative on Antimicrobial Resistance (JPIAMR), initiated by Sweden and Italy and aimed at coordinating participating countries’ research in the area for a more efficient use of resources and stronger action towards these challenges. The plan also states the responsible actors for each area of interest.

#### 3.1.4. Swedish Work on Containment of Antibiotic Resistance

This is a report of Sweden’s work towards rational use of antibiotics and improved resistance surveillance. It was developed by a Swedish team of experts within the collaboration between the Indian National Centre for Disease Control and the Public Health Agency of Sweden. It focuses on human medicine and aims to provide a toolbox of previous Swedish experiences that can be adapted to other countries’ needs.

The report highlights multidisciplinary collaboration and contains an extensive description of the Swedish strategy for containing antibiotic resistance. This includes resistance monitoring, antibiotic consumption monitoring, evidence-based treatment recommendations to support prescribers, feedback systems to prescribers about their antibiotic prescribing, the creation of Strama (the Swedish strategic programme against antibiotic resistance, a national level voluntary network of agencies and organisations) and its work, the SWEDRES/SVARM yearly report (A Report on Swedish Antibiotic Utilisation and Resistance in Human Medicine (SWEDRES) and Swedish Veterinary Antimicrobial Resistance Monitoring (SVARM)), and target groups for communicating rational use of antibiotics.

#### 3.1.5. Surveillance of Antibiotic Resistance: A National Plan

This plan is mainly aimed at actors within clinical microbiology, infection control, Strama and infection prevention, and control in healthcare facilities. It discusses general principles of surveillance of antibiotic resistance and describes current common systems for antibiotic resistance surveillance in Sweden (Svebar, SmiNet, and ResNet) and the European Antimicrobial Resistance Surveillance (EARSnet). Then, it presents the different antibiotic resistance surveillance plans at local and national levels and how this links to EARSnet. The plan focuses on human data but mentions that these should be analysed together with resistance data from the other sectors (food, environment, animals).

### 3.2. Context

In the following table we present a timeline of important events that, in various ways, have contributed to the Swedish work for containing antibiotic resistance (Table 1). This timeline was based on the analysed documents and gives an overview of the context in which the policies were developed. The table also includes some leading international efforts for comparison.

Structural factors in the Swedish healthcare system as early as 1959 have played an important role in Sweden’s work against antimicrobial resistance, driven by events such as the large Salmonella outbreak in 1953. Thousands of human cases and 90 deaths were recorded and the outbreak could be traced to infected meat from a slaughterhouse in Alvesta [26]. In the early 1990s, there was a rapid increase in pneumococcal strains resistant to penicillin in southern Sweden. This triggered the foundation of the Swedish strategic programme against antibiotic resistance, Strama, in 1995. Even earlier, the animal health community began work in the field and antibiotics as growth promoters for animals were prohibited in Sweden in 1986, 20 years ahead of the EU.

The term “One Health” was first used in 2003–2004 after several severe outbreaks of zoonotic diseases, for example, SARS, Avian Flu, and West Nile Virus [27]. Later, a Global Action Plan on antimicrobial resistance [6] was developed by the tripartite alliance (WHO, World Organization for Animal Health (OIE) and the Food and Agriculture Organization (FAO)) and endorsed at the UN general assembly meeting on antimicrobial resistance in 2016. This was followed by the EU commission action plan for antimicrobial resistance (AMR) from a One Health perspective in 2017.

From an international perspective, the level of antibiotic use and the prevalence of resistant bacteria in animals in Sweden is low. According to a government decision in 2010 (S2010/7655/FS), the National Board of Health and Welfare in collaboration with the Swedish board of agriculture, should develop an intersectoral action plan for coordinated work against antibiotic resistance and healthcare-associated infections (HAI). The responsibilities of the National Board of Health and Welfare were transitioned to the Public Health Agency of Sweden on 1 July 2015. The global commitments made within the 2030 Agenda, as well as Sweden’s Policy for Global Development [28], are key frameworks for Sweden, aiming for continued international leadership in the work of containing antibiotic resistance.

### 3.3. Process

Sweden has had a strong political commitment to the work to contain antibiotic resistance ever since the problem became evident in the 1990s. Strama has been a key driver in the work against antibiotic resistance with early work in the subsections for veterinary and dental care. Strama was formed in 1995 as a voluntary network at the national and regional levels, with local groups. In 2000, Strama began receiving government funding. Initially, the national level (the steering group) was an independent organisation with a working group consisting of members from authorities and organisations. It was formalised and incorporated into the Public Health Agency of Sweden in 2010. The regional level (the Strama Network) remains independent from the Agency but receives funding from the regions.

Swedish stakeholders see the interdisciplinary collaboration in Sweden, as a role model globally and international collaboration has been key in Sweden’s strategy. The policies aim to intensify the work. As stated by the authorising ministers in the most recent document (key document No. 1), “we cannot wait, we must act now”.

National health policies are developed by the Swedish Parliament, and the Ministry of Health and Social Affairs is responsible for achieving the objectives. Implementation of the policies is supported by a number of national and regional actors, as shown in Figure 1.

### 3.4. Actors

A vast range of actors are mentioned as important in the Swedish work to contain antibiotic resistance. Key document No. 1, “The Swedish strategy to combat antibiotic resistance” states, “action is needed both from academic research and industry if new antibiotics, vaccines and diagnostic methods are to be developed”. Focusing on a One Health perspective, the Swedish strategy is in line with the Global Action Plan (GAP) developed by WHO, FAO, and OIE. Consequently, the following Swedish ministers have joined forces to form the strategy: The Minister for Health and Social affair, the Minister for Rural affairs, the Minister for Higher education and research, the Minister for International Development Cooperation, and the Minister for Environment and Climate.

Several authorities support the Ministries’ activities. The main authorities in human medicine, according to key document No. 4, are the National Board of Health and Welfare, the Public Health Agency of Sweden, and the Medical Products Agency. There are also specialist associations such as the Swedish Association of Local Authorities and Regions (SALAR). At the regional and local level, microbiological laboratories, communicable disease units, Strama groups, pharmaceutical committees, and infection control units are involved. Every region has a regional medical officer for communicable disease control who is responsible for planning and leading the regional work for communicable disease prevention and control. In addition, Sweden has a strong core of infectious disease specialists providing in-depth knowledge of transmission of infectious diseases, as well as veterinary practitioners with a high level of awareness of the ecological implications of increasing resistance.

As mentioned, Strama, the Swedish strategic programme against antibiotic resistance, has been a key driver and was formed in 1995 by the following parties: The Swedish Society of Medicine’s expert group on antibiotic issues, the Swedish Institute for Communicable Disease Control; the Swedish Society for Communicable Disease Prevention and Control; the Medical Products Agency; the National Board of Health and Welfare; Apoteksbolaget AB (The National Corporation of Swedish Pharmacies, the Swedish state owned pharmaceuticals retailer); NEPI, the Network for Pharmaceutical Epidemiology; the National Veterinary Institute; and representatives of drug and therapeutic committees (called “pharmaceutical committees” in Figure 1). Strama was formed as a voluntary professional network and, from its inception, was chaired by Professor Otto Cars, who remained its chair from 1995 to 2011. Professor Cars is an infectious disease specialist with a strong engagement and was extremely influential in the Swedish work against antibiotic resistance. He was also instrumental for the inception of the ReAct network in 2005.

In the key document No. 2 “Revised multisectoral action plan against antibiotic resistance 2018–2020”, the government of Sweden has given the responsibility of coordinating the multisectoral work against antimicrobial resistance to the Public Health Agency of Sweden and the Swedish Board of Agriculture. Twenty-five national agencies and organisations from different societal sectors are included in the multisectoral work (Appendix B). These are the same agencies and organisations that have collaborated in the development of the “Action plan against antibiotic resistance and healthcare associated infections—background for the agencies’ continued work” (key document No. 3). The action plan only concerns the national agencies and boards involved in the work, but not stakeholders that will do the work “on the ground”.

## 4. Discussion

This paper analyses the Swedish policies for containment of antibiotic resistance. Sweden has worked at the policy level since the 1980s to promote rational use of antibiotics and contain antibiotic resistance. From an international perspective, Sweden started developing and implementing these policies early. Sweden’s success in preventing, containing, and mitigating antibiotic resistance may be attributed to a strong underpinning policy framework that is integrated into all relevant governmental systems, and is supported by adequate human, infrastructural, and operational resources in all sectors of the One Health triad. The Swedish work has also been driven actively by practitioners on the ground as well as local role models and champions such as Professor Otto Cars.

The Swedish policies have a clear One Heath perspective. This is evident in the content, the actors, and the process in which the Swedish strategy focuses on intersectoral collaboration and includes human and veterinary medicine, agriculture, food production, and environmental sectors. However, a limitation in the reporting on AMR is seen in the yearly Swedres-Svarm report on antibiotic consumption and resistance which reports on human and animal health and does not include antibiotics in the environment. Furthermore, although the report has a chapter comparing veterinary and human antibiotic sales and resistance, there is no attempt to follow a more One Health integrated approach [19]. The Swedish policies state international collaboration as key, very much in line with the World Health Assembly resolution and Global Action Plan on Antimicrobial Resistance [6,29]. Antibiotic resistance is considered to be one of the most complex health challenges today, and it is only possible to tackle it through a concerted global effort [30]. As part of these efforts, better coordination, and thereby synergies between innovators, researchers, funders, and policy makers are needed [31], and this is also explicitly stated in the Swedish policies.

Stakeholders in Sweden see Sweden as a model country when it comes to containment of antibiotic resistance, referring to the low use of antibiotics and well-established policies to support this. With the global spread of resistance, Sweden’s favourable situation will be affected by actions taken outside its borders. The Swedish strategy, therefore, aims for Sweden to show leadership internationally through cooperation in various fora, such as the EU, OIE, WHO, FAO, and OECD. The One Health concept and AMR has rapidly reached the political agenda globally, but policy development and implementation in the field in low- and middle-income countries is often lagging behind and needs special consideration [32,33]. To be able to implement the “Swedish model” in other countries, a clear set of adaptable process and outcome indicators need to be developed using an implementation research framework. Implementation research is defined as “the scientific inquiry into questions concerning implementation—the act of carrying an intention into effect, which in health research can be policies, programmes, or individual practices (collectively called interventions).” It focuses on “what, why, and how interventions work in “real world” settings” including local context and stakeholders to ensure “acceptability, adoption, appropriateness, feasibility, fidelity, implementation cost, coverage, and sustainability” of the interventions [34]. It is worth mentioning the structural factors in Sweden that may have contributed favourably to the successful implementation of AMR policies, for example, a tax-funded health insurance for all residents in the country, the ability to stay at at home for a week without a sick leave certificate, and paid leave to stay at home with sick children. In addition, the country has a high level of education in the population enabling awareness raising, and a comparatively good hygiene standard with a low infectious disease burden.

A challenge with policies is achieving the wanted effects when implementing them in practice. Evaluating the effect of the policies is not the scope of this paper, but data show that Sweden has low and decreasing consumption of antibiotics for both human and veterinary use in an international comparison [18,19,20,26]. Qualitative studies conducted as part of the ABR CARRO project by our research team have shown that actors both in the human, veterinary, and environmental sectors perceive the Swedish work on AMR to be well known and established among most stakeholders in the various sectors [35,36,37]. Most stakeholders feel that a lot has been achieved in Sweden regarding antibiotic resistance containment but that there is still room for increasing awareness and emphasising changes in behaviour and attitudes among prescribers and the public. With regards to the term “One Health” itself, knowledge among stakeholders was relatively low [35,36,37].

A study of the One Health concept and its effect on antimicrobial resistance (AMR) policies in the UK and Australia shows that the UK policies mentions the concept less than the Australian equivalent [38]. However, this is not reflected in their actions to contain AMR, where Australia has much less surveillance of antibiotic use outside the human sector [38]. We conclude that using the One Health concept might be useful for agenda setting but that in policy making, other factors, such as the way a country’s institutions work (centralised government in the UK vs. multilevel federal system in Australia), are more important for success and that “forcing” policies to fit one concept might lead to ignoring actionable ideas [38]. In Sweden, the policies to contain AMR are made at the central level but implemented by local actors. A recent study of 77 national action plans on AMR from a range of countries at various income levels found that although almost all of them address One Health, the measures are applied mainly in the human sector [39]. Similarly, a literature review regarding design and implementation of One Health approaches showed that the greatest knowledge gap was for monitoring and evaluating the One Health initiatives [40]. More research, including evaluating implementation effect, is needed to understand if and how a One Health focus can be used to help organisations and actors take effective measures to combat antibiotic resistance.

## 5. Conclusions

Sweden has had a long tradition of work for containing antibiotic resistance. The work started in separate sectors, but since 2000, there are policies in place with a clear One Health focus. The policies have clear aims with specific action points and responsible actors. International collaboration is key in the policies and the Swedish government aims for Sweden to be a model and to show leadership in the work to contain antibiotic resistance.

## Figures and Tables

**Figure 1 antibiotics-10-00526-f001:**
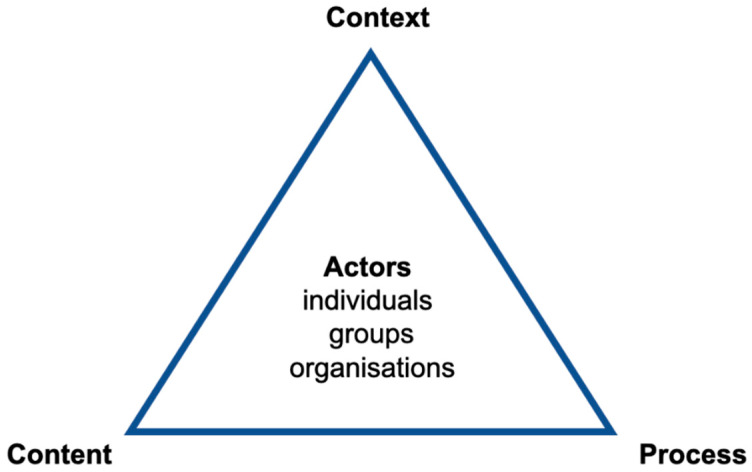
The policy analysis triangle framework, by Walt and Gilson [22].

**Table 1 antibiotics-10-00526-t001:** Timeline of events important for the Swedish work for containing antibiotic resistance.

Year	Event
1959	Recommendations for preventing spread of infections in healthcare (by “Medicinalstyrelsen”, the precursor of the Swedish National Board of Health and Welfare)
1960s	First physicians and nurses specialised in disease prevention and control employed in Swedish healthcare
1980s	Alcohol-based hand disinfection recommended in healthcare in Sweden
1986	Use of antibiotics as growth promotors prohibited in Sweden (at the time an action from farmers reacting to customers not wanting meat with antibiotic residues, rather than an action to combat antibiotic resistance)
1989	System of regional medical officers for communicable disease control established
1995	Strama (Swedish strategic programme against antibiotic resistance) Formed as a voluntary network of agencies and organisations at a national level
2000	SPAR plan (National action plan for containment of AMR), an intersectoral action plan
2001	WHO strategy for containment of AMR and EU resolution that recommended each member state to create intersectoral steering groups for AMR containment
2002	First Swedish Antibiotic Sales and Resistance in Human Medicine (SWEDRES) report published on antibiotic prescriptions and resistance
2003–2004	The term “One Health” is first used internationally, in association with the SARS epidemic
2003	SWEDRES report published in collaboration with Swedish Veterinary Antimicrobial Resistance Monitoring (SVARM)
2005	ReAct (Action on Antibiotic Resistance) is created by representatives from Sweden, i.e., Strama, Dag Hammarskjölds Foundation, and Karolinska Institutet, and a group of international representatives
2006	Use of antibiotics as growth promotors prohibited in EU (after suggestion by Sweden)
2006	Strategy for coordinated work towards the containment of antibiotic resistance and healthcare-related diseases
2006	Rules put in place to ensure that all providers of human healthcare have access to competence in infection control
2007	The National Board of Health and Welfare’s regulation about basic hygiene in health care becomes effective
2007	Strama dental care formed
2008	Strama VL (veterinary and foodstuff) formed
2010	Joint Programming Initiative on Antimicrobial Resistance Top of Form
	Search (JPIAMR) founded (Sweden co-initiator)
2010	4-year patient safety initiative by the Swedish Association of Local Authorities and Regions, in which rational use of antibiotics and healthcare associated infections (HAI) are central
2011–2014	The government and the Swedish association of local authorities and regions carry out a patient safety drive where containment of AMR is one of 4 targets, the drive contributes to reduced antibiotic prescribing in primary care
2011	The National Board of Health and Welfare suggests ways of developing the strategy to contain ABR and HAI, together with other authorities
2012	Intersectoral coordinating mechanism established for antibiotic resistance and HAI; the National Board of Health and Welfare and Swedish Board of Agriculture jointly responsible and 21 government agencies participate
2013	Veterinarians’ right to prescribe antibiotics meant for humans is limited, to reduce resistance development and correct antibiotics are prescribed for animals
2013	Group formed made up of actors working with infection control in human medicine at a national level
2014	Work for a renewed plan for the Swedish work against antibiotic resistance and hospital acquired infections appointed to the intersectoral coordinating mechanism
2014	The Swedish board of agriculture’s regulation for hygiene plans to prevent infections in animal healthcare comes into effect
2014	Sweden hosted a global consultation, which resulted in advice on preparation of a manual on early implementation of the Global Antimicrobial Resistance Surveillance System (GLASS); the Public Health Agency of Sweden is now the WHO Collaborating Centre for antimicrobial resistance containment
2015	Global Action Plan on Antimicrobial Resistance endorsed by the World Health Assembly
2015	Strama programme council (“Programråd Strama”) is established within the Swedish association of Local authorities and regions as part of the work on knowledge governance
2016	Swedish strategy to combat antibiotic resistance
2017	The government renews the commission for national, intersectoral collaboration, and a revised action plan against antibiotic resistance, with 25 agencies and organisations participating.
2020	Updated Swedish strategy to combat antibiotic resistance

## Data Availability

No new data were created or analysed in this study. Data sharing is not applicable to this article.

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
