# Peer review of "Exploring the One Health Perspective in Sweden’s Policies for Containing Antibiotic Resistance"

_antibiotics, 2021, doi:10.3390/antibiotics10050526_

Round 1
Reviewer 1 Report
In this manuscript, the authors, Eriksen et al., present an interesting policy analysis on Sweden's multi-sectoral strategic program to curtail antibiotic resistance in this country. The methodology is sound and clearly presented, and the content is within the scope of the journal Antibiotics. This work should be of general interest to readers concerned or involved in antibiotic stewardship efforts. Minor comments/suggestions are provided as follows:
(1) Briefly describe what Strama is when first introduced (line 204).
(2) Define the abbreviation ESBL -- e.g., ESBL (extended-spectrum beta-lactamases) (line 173).
(3) Please consider removing unnecessary hyphens.
(4) Line 99: Semicolon should be changed to a colon.
(5) Line 356: Remove "policies" right before "AMR policies".
(6) Lines 359 and 360: Please insert a verb in the sentence to make it complete.
Author Response
Thank you for this positive feedback. Below follow responses to your specific comments:
- Briefly describe what Strama is when first introduced (line 204).
Response: The following has now been added to describe Strama: “(the Swedish strategic programme against antibiotic resistance - a national level voluntary network of agencies and organisations)”
- Define the abbreviation ESBL -- e.g., ESBL (extended-spectrum beta-lactamases) (line 173).
Response: The definition has now been added
- Please consider removing unnecessary hyphens.
Response: We have now removed hyphens throughout the manuscript from several words, e.g. multifactorial, multidisciplinary and transdisciplinary
- Line 99: Semicolon should be changed to a colon.
Response: This has been changed
- Line 356: Remove "policies" right before "AMR policies".
Response: The extra “policies” has been deleted.
- Lines 359 and 360: Please insert a verb in the sentence to make it complete.
Response: Thank you for noticing this omission. The sentence has now been changed to: “In addition, the county has a high level of education in the population enabling awareness raising, and a comparatively good hygiene standard with a low infectious disease burden.”
Reviewer 2 Report
The authors investigated the regulations to prevent antibiotic resistance in Sweden. Despite the authors focused only on Sweden, their observations are also important for the general population as it indicates the possible directions of policies needed to be undertaken in other countries as well to prevent antibiotic resistance. The paper is well written and provides in-depth analysis making it a valuable report.
Author Response
Thank you for this encouraging comment.
Reviewer 3 Report
very little changes needed. important subject, to be sure. i'd like to have a more 'in depth' discussion of one or two other countries who've attempted to grapple with this issue. perhaps japan or south korea?
Author Response
Thank you for this comment. The authors are aware that many countries have include a One Health perspective in their AMR policies, as we have included in our discussion. We are also aware of Japan's National Action Plan on antimicrobial resistance 2016–2020 which has a clear One Health approach and was developed through multisectoral collaboration. However, the current paper is a study of the One Health perspective in Sweden’s policies. The authors therefore feel that including an in-depth discussion of other countries would change the topic of the paper and would require a major revision.